# Spectrum and Clinical Interpretation of *TTN* Variants in Ecuadorian Patients with Heart Disease: Insights into VUS and Likely Pathogenic Variants

**DOI:** 10.3390/ijms262411896

**Published:** 2025-12-10

**Authors:** Patricia Guevara-Ramírez, Santiago Cadena-Ullauri, Rafael Tamayo-Trujillo, Viviana A. Ruiz-Pozo, Elius Paz-Cruz, Rita Ibarra-Castillo, José Luis Laso-Bayas, Ana Karina Zambrano

**Affiliations:** 1Universidad UTE, Facultad de Ciencias de la Salud Eugenio Espejo, Centro de Investigación Genética y Genómica, Quito 170129, Ecuador; 2Independent Researcher, Quito 170103, Ecuador

**Keywords:** cardiovascular disease, genome, precision medicine, NGS

## Abstract

This study described *TTN* gene variants in Ecuadorian patients with hereditary cardiac diseases, integrating genetic ancestry to improve variant interpretation in an underrepresented population. Sixty patients with confirmed hereditary cardiac conditions were analyzed using the TruSight Cardio NGS panel (Illumina, San Diego, CA, USA), which targets 174 cardiac-associated genes. Bioinformatic analyses and classification were performed in accordance with ACMG/AMP guidelines, and ancestry inference was conducted using 46 Ancestry Informative Markers (AIM-InDels). From 4008 detected *TTN* variants, 29 variants of interest remained after filtering: 27 classified as variants of uncertain significance (VUS) and two as likely pathogenic. All variants were heterozygous and distributed across exons 3–358, primarily in the A-band region, commonly associated with cardiomyopathies and arrhythmic phenotypes. Two truncating variants (exons 267 and 272) met PVS1 criteria, while several missense variants (p.Ser91Gly, p.Pro12140Ser, p.Arg34653Cys) showed possible modulatory effects on hypertrophic or arrhythmic outcomes. Genetic ancestry revealed a predominant Native American background, followed by European and African components. These findings expand the understanding of *TTN*-related cardiac disease in Latin America, suggesting that *TTN* functions as a genetic modifier influencing disease expression. Incorporating ancestry information enhances genomic interpretation and supports precision medicine in diverse populations.

## 1. Introduction

Cardiovascular diseases (CVD) represent a major global health burden and account for approximately 29% of annual deaths in Latin America and the Caribbean [1]. Within this broad category, inherited cardiomyopathies and cardiac arrhythmias represent major contributors to morbidity and sudden cardiac death due to their significant clinical impact and underlying genetic basis [2].

Genetic cardiac arrhythmias comprise a group of disorders characterized by abnormal heart rhythm, which may be caused by mutations in genes that encode or regulate ion channels involved in excitability and electrical conduction [3]. These include primary channelopathies, such as long QT syndrome, Brugada syndrome, and atrial fibrillation, as well as cardiomyopathies associated with an increased risk of arrhythmia [4].

Furthermore, cardiomyopathies are defined by structural and functional abnormalities of the myocardium [5]. Major subtypes include arrhythmogenic right ventricular cardiomyopathy (ARVC), dilated cardiomyopathy (DCM), hypertrophic cardiomyopathy (HCM), and restrictive cardiomyopathy (RCM). Their prevalence ranges from 0.004% to 0.2%, and familial forms account for approximately 20–50% of all cases [6].

In this context, variants in genes such as *KCNQ1*, *KCNH2*, *SCN5A*, *MYPN*, and *TTN* have been associated with various cardiac phenotypes [7,8]. Notably, structural genes such as *TTN* are often linked to cardiomyopathies and have also been associated with arrhythmic phenotypes, suggesting a notable connection between structural and electrical heart diseases [9].

The *TTN* gene is composed of 364 exons and encodes titin, the largest known human protein. Titin plays crucial structural and signaling roles in striated muscle [10]. The protein spans across the Z-disk, I-band, A-band, and M-line regions of the cardiac sarcomere [11]. The modular organization of the protein influences its mechanical function and vulnerability to mutations. Furthermore, titin is fundamental to sarcomere organization and myocardial elasticity, underscoring its role in maintaining cardiac contractile function [12].

The use of Next-Generation Sequencing (NGS) technologies has enabled the identification of *TTN* variants and their association with diverse cardiac conditions, including cardiomyopathies, conduction system diseases, and primary electrical disorders [13].

Moreover, a population’s ancestry may influence the frequency of genetic variants associated with CVDs. Latin American populations, including those of Ecuador, exhibit a mixture of ancestral components from Native Americans, Europeans, and Africans [14]. This ancestral genetic heterogeneity has significant implications for the genomics of heart disorders, as it may reveal specific genetic variants not reported in genomic databases [15].

Thus, this study describes genetic variants in the *TTN* gene of Ecuadorian patients with heart disease, incorporating analysis of individual ancestral components. These findings provide valuable insights into the genetic architecture of cardiovascular disease in the Ecuadorian population and contribute to the broader understanding of *TTN*-associated cardiac disorders in underrepresented populations.

## 2. Results

A total of 4008 variants were initially identified in the *TTN* gene among the 60 Ecuadorian patients. After applying quality control filtering, 3116 high-quality variants were retained for further analysis. Subsequent filtering based on predicted functional consequence excluded synonymous and intronic variants, leaving 1597 variants with potential functional impact.

After removing variants annotated as benign or likely benign in ClinVar, a final set of 29 variants in 21 patients was selected for detailed evaluation (Table 1). According to ACMG/AMP classification, 27 *TTN* variants were classified as VUS, and two as likely pathogenic. All variants were mapped to the canonical *TTN* transcript NM_001267550.2 and distributed across exons, ranging from exon 3 to exon 358 (Figure 1).

*TTN* variants were detected in patients with various cardiac phenotypes, as shown in Figure 2. The highest frequencies were observed in hypertrophic cardiomyopathy and long QT syndrome, followed by bradycardia, atrioventricular block, and dilated cardiomyopathy. Fewer variants were detected in atrial fibrillation, arrhythmogenic cardiomyopathy, and Wolff–Parkinson–White syndrome.

Of the 29 variants analyzed, 27 (93%) were missense substitutions, while two (7%) were nonsense variants introducing premature stop codons in exons 267 and 272. All variants were heterozygous. According to ACMG/AMP evidence codes, all 29 variants met PM2 criteria, consistent with moderate pathogenic evidence due to their low frequency in gnomAD. Similarly, five variants met PP3 criteria, reflecting supporting in silico pathogenicity, and two variants were classified as PVS1, representing very strong evidence for pathogenicity due to predicted loss of function in a gene with an established disease mechanism. Conversely, five variants met BP4 criteria, indicating that in-silico analyses support a benign role, and four variants were classified as BP6, suggesting a benign role based on reputable sources.

Additionally, cross-referencing with ClinVar revealed that 19 of the 29 variants (65.5%) had no prior record, and several lacked rsID assignment, which may suggest rarity and potential population specificity.

### Genetic Ancestry Determination

Genetic ancestry analyses revealed a predominantly Native American (NAM) genetic background among the present cohort. The average ancestry composition across the 20 genotyped individuals was 63.1% NAM, 29.1% European (EUR), and 7.7% African (AFR). Individual ancestry proportions are summarized in Appendix A.

Notably, NAM ancestry exceeded 70% in ten participants, while a mixed NAM and EUR ancestry, ranging from 40% to 60% was observed in seven individuals. Only one participant exhibited a markedly higher AFR component (53.3%). These results highlight the complex admixture found in the Ecuadorian population. Figure 3 shows a triangle plot, displaying the clustering between Native American and European reference populations.

## 3. Discussion

In the present study, 29 *TTN* genetic variants were identified in our cohort of Ecuadorian patients with hereditary heart disease. The variants were distributed across exons 3 to 358, covering regions from the N-terminal portion (Z/I-band) to intermediate and distal domains (A-band). The variants located between exons 3 and 49, corresponding to the Z disc and Z/I junction, map to regions that anchor titin to α-actinin, myopalladin, and telethonin. Samples ADN116 (exon 3) and ADN150 (exon 49) carried variants in this area, associated with hypertrophic cardiomyopathy and Ebstein’s anomaly, respectively. These regions include the Z1–Z2 and Z-repeats domains, which are essential for the formation of highly stable mechanical complexes between TTN–α-actinin–teletonin. Although these portions are not traditionally associated with direct contractile dysfunction, alterations in the Z domains can disrupt Z-disc assembly and hypertrophic signaling mediated by the MLP–TCAP–TTN complexes, which are involved in the myocardium’s adaptive response to mechanical overload [5].

Most variants in this cohort were located between exons 244 and 358, corresponding to the A-band, a domain frequently associated with pathogenic phenotypes [16]. The A-band contains key myosin-binding elements, including sites for sarcomeric myosin-binding protein C (MyBP-C), which interact with the myosin tail to stabilize the thick filament and support force generation during contraction [5]. Experimental evidence has shown that homozygous *TTN* truncation variants (*TTN*tvs) in the C-terminal region (exon 326) of the *TTN* gene can lead to fetal death in a knock-in mouse model [17], and worsen DCM phenotypes [18]. In our cohort, several individuals harbored missense variants in the C-terminal region (exons 325, 326, 344, and 358) of the A-band. These missense variants may alter interactions with sarcomeric proteins and thereby contribute to cardiomyopathic phenotypes such as DCM.

Variants were also identified in exon 326, although only heterozygous missense changes were detected (Table 1). These diverse clinical phenotypes could be related to inter-individual differences in the expression of the wild-type *TTN* allele, which may potentially interfere with sarcomere assembly and its interaction with non-sarcomeric proteins. Experimental evidence further supports the functional relevance of this region. For instance, a mouse model embryo carrying a homozygous *TTN* knock-in mutation in exon 326 (c.43628insAT, p.Ser14450fsX4) showed defects in sarcomere formation, which led to a lethal phenotype. Moreover, in the same study, the authors studied the variant in heterozygosis, which was associated with ventricular dilatation and upregulation of the wild-type titin allele, leading to diffuse myocardial fibrosis under angiotensin II exposure [17].

Additionally, 10 *TTN* variants were identified in exons 253, 267, 272, 284, 287, 313, 315, 325, and 344 of the A-band were also detected (Table 1). Seven were missense variants, two were stop-gained, and one affected a splice region. *TTN* missense variants have been associated with impaired titin, leading to defects in sarcomere assembly and function [5,19,20]. However, the role of missense variants in the severity of cardiomyopathies is modest, mainly due to the absence of clinical differences between DCM individuals with and without *TTN* missense variants [21]. The presence of DCM phenotypes in individuals harboring *TTN* missense variants may be explained by alterations in protein folding, which could disrupt titin function [22]. In this context, considering that exons of A-band are highly expressed regions due to their high variable percentage spliced-in (PSI) values, missense variants could alter titin structure at the A-band level, which may influence its interaction with non-sarcomeric proteins, such as myosin [5,23]. Therefore, understanding the impact of *TTN* missense variants on protein folding and their interaction with sarcomeric and non-sarcomeric proteins could enhance our understanding of DCM pathogenesis.

Two *TTN*tvs, associated with DCM and cardiac arrhythmia, were identified in the present study (Table 1). These truncating variants are located in exons 267 and 272 of the *TTN* A-band. Several studies have described the role of *TTN*tvs in the A-band region in the development of DCM [5,23,24,25,26]. These *TTN*tvs have been related to defects in alternative splicing, leading to an altered protein structure of the cardiac isoforms N2B and N2BA [5]. Such truncated exons could disrupt protein assembly and its interaction with sarcomeric and non-sarcomeric proteins [22]. Moreover, these TTNtvs could promote the Nonsense-Mediated mRNA Decay pathway of the mutated allele, ultimately altering protein structure [27]. The epigenetic modifications of mutated and wild-type alleles could also be involved in cardiomyopathies in the presence of *TTN*tvs [28]. *TTN*tvs play a crucial role in DCM through their effects on sarcomere structure and function. Thus, assessing novel *TTN*tvs in animal models could help clarity the underlying pathophysiology and guide therapeutical development.

Among the 29 TTN variants identified, seven heterozygous missense variants (p.Leu24712Pro, p.Pro18399Ser, p.Arg14996Cys, p.Arg34276Gln, p.Glu31822Lys, p.Ser91Gly, and p.Val15833Glu) were specifically observed in patients diagnosed with HCM (Table 1). These variants, located between exons 3 and 358, were all classified as PM2 due to their absence or very low frequency in population databases, and three are listed in ClinVar as VUS. Previous studies have reported that the frequency of *TTN* variants in HCM and in healthy individuals is comparable (2–3%), initially raising questions about their pathogenic relevance [29,30]. However, a more in-depth analysis of a cohort of 529 Chinese patients with HCM confirmed that, although the prevalence of *TTN*tvs did not differ significantly from controls (2.5% vs. 2.6%), their presence was associated with an increased risk of adverse cardiovascular outcomes, including cardiac death [31]. These findings support the hypothesis that *TTN* does not act as a primary causal gene in HCM, but rather as a genetic modifier which may influence the severity and clinical progression of the disease [32].

Within this HCM-associated subset, the p.Ser91Gly (exon 3) variant is notable for its location in the Z-disc domain, a structurally and functionally important region of titin. This same domain also contains the p.Arg740Leu mutation reported by Satoh et al. (1999) in familial HCM [33]. Although these variants originate from different cohorts, their shared localization in a key Z-disc region, where titin interacts with proteins such as α-actinin, telethonin (TCAP), and MLP, underscores the functional importance of this domain and highlights the need for further investigation of variants occurring in this area [5,34].

The remaining HCM-associated variants (p.Leu24712Pro, p.Pro18399Ser, p.Arg14996Cys, p.Arg34276Gln, p.Glu31822Lys, and p.Val15833Glu) are located in exons 244, 253, 284, 326, 344, and 358, which correspond to the I, A, and M-band regions of *TTN*. These exons show high myocardial inclusion (hiPSI) and constitutive expression, as described by Schiabor Barrett et al. (2023) [35], and encode structural domains essential for sarcomeric integrity and contractility through interactions with myosin and MyBPC3 [36]. The presence of HCM-associated variants within these structurally critical regions supports the hypothesis that variants in hiPSI exons may modulate sarcomeric performance and contribute to clinical variability observed among *TTN*-related hypertrophic phenotypes.

On the other hand, long QT syndrome (LQTS) is characterized by QTc prolongation, ventricular arrhythmias, and an increased risk of sudden cardiac death [37]. While traditionally associated with ion channel mutations, the genetic spectrum of LQTS has expanded to include structural and sarcomeric genes such as *TTN* [38,39]. In this cohort, three heterozygous missense variants in *TTN* were identified in patients with LQTS: p.Arg34653Cys and p.His33987Tyr (exon 358) and p.Pro12140Ser (exon 170). All were assigned with the PM2 criterion due to their rarity in population databases. Additionally, p.Pro12140Ser also met the BP4 criterion, based on bioinformatic predictions suggesting a likely benign effect.

One of the cases analyzed (ADN001) was previously reported by our group in a study of channelopathies, which described the coexistence of mutations in *KCNH2*, *AKAP9*, and the *TTN* p.Arg34653Cys variant, now reevaluated in this study. In that report, the patient presented an LQT2 phenotype with T-wave abnormalities and recurrent arrhythmic events, suggesting a modulatory effect of TTN that could potentiate electrical dysfunction when variants in potassium channel genes coexist [38]. Consistently, recent studies indicate that rare missense variants in *TTN* (*TTN*mvs) can induce ionic remodeling and QT prolongation, even without obvious structural alteration, supporting their role in electrical instability and the clinical severity of arrhythmic syndromes [40].

Although most population and familial reports have focused on *TTN*tv, evidence supports a broader role for *TTN* in cardiac electrical regulation. For instance, in the ARIC (Atherosclerosis Risk in Communities) study, which analyzed more than 6000 exomes, 27 rare nonsynonymous variants associated with QT interval variation were identified, 22 of which corresponded to *TTN* [41]. Collectively, these findings reinforce the emerging role of *TTN* as a modulator of arrhythmic risk, encompassing both truncating and missense variants, and highlight the need for functional validation studies to clarify its role in the pathophysiology of LQTS.

Finally, we also identified heterozygous missense variants in the *TTN* gene in patients with cardiac conduction and rhythm disorders, including atrial flutter, atrial fibrillation, supraventricular tachycardia, atrioventricular block, bradycardia, and Ebstein’s anomaly (Table 1). Most variants met the PM2 criterion, as they were absent or had a low frequency in population databases. Variants such as p.(Pro12140Ser), p.(Val12094Ala), and p.(Val13084Met) also met BP4, with in silico predictions supporting a likely benign effect. In contrast, p.(Pro21891Leu), p.(Tyr24979His), and p.(Gly4768Ser) showed a PP3 profile, consistent with a possible functional impact according to evolutionary and structural models, suggesting that certain regions of *TTN* could be involved in myocardial electrical phenotypes. These findings align with recent evidence showing that *TTN* variation contributes to electrical phenotypes, particularly early-onset atrial fibrillation [42] and familial conduction disease [43]. Large cohort studies have reported rare *TTN* variants enriched in individuals with arrhythmic traits and associated with more persistent AF forms, systolic dysfunction, or progressive conduction abnormalities, supporting a modulatory rather than primary pathogenic role for *TTN* in electrical instability [44,45]. In this context, the variants observed in our cohort may reflect a similar modifier effect, although functional and segregation studies will be required to clarify their contribution to conduction system disorders.

Beyond genotype–phenotype correlations, demographic features in this cohort provide additional context for *TTN* variant interpretation. The sex distribution showed a higher proportion of men than women, consistent with global reports describing increased prevalence and severity of cardiomyopathies and hereditary arrhythmias in males [46]. Experimental and clinical data suggest that male sex hormones can promote hypertrophic remodeling, fibrosis, and oxidative stress, whereas estrogens exert cardioprotective effects by modulating survival and anti-inflammatory pathways [47,48,49,50]. Collectively, these factors could contribute to the earlier and more severe presentation of cardiomyopathic phenotypes in males, reinforcing the importance of considering biological sex in the clinical and prognostic interpretation of *TTN* variants in hereditary heart disease.

In relation to age, *TTN* variants were identified across a wide range of ages from newborns to older adults, with the highest frequencies between 11 and 20 years and between 41 to 50 years of age. In early childhood (0–10 years), the detection of *TTN* variants suggests a congenital or early familial onset, possibly associated with severe or syndromic forms of cardiac dysfunction. Conversely, the higher burden of variants observed in middle age (41–50 years) could be related to the cumulative interaction between genetic predisposition and acquired factors, such as hypertension, hemodynamic stress, or environmental exposures, which can trigger the clinical expression of pathogenic variants or VUS.

The genetic ancestry analyses showed a predominant NAM genetic background in the present cohort, followed by EUR and AFR components, consistent with previously published results [51]. When ancestry was considered together with the *TTN* variant profiles, several differences emerged relative to the populations in which these variants have been previously reported. For instance, the *TTN* variant detected in sample ADN023, which has 90.7% NAM ancestry, has been reported only in African, European, Japanese, and African American cohorts [52,53,54,55,56,57,58]. Similarly, the variants identified in samples AND099, ADN117, and ADN120, all with NAM ancestry levels above 90%, have been described exclusively in European populations [52,53]. These discrepancies may be explained by the underrepresentation of Latin American and Indigenous populations in global reference databases, resulting in a potential misestimation of variant frequency and uncertainty in classification [59,60,61].

Interestingly, the variant identified in sample ADN116 and the two variants detected in sample ADN146 have not been reported in any population and lack an rs identifier [52,53,57,58]. These findings suggest the presence of novel or rare alleles within the Ecuadorian population and further highlight the scarcity of regional genomic data in public repositories.

The observed ancestry distribution provides a valuable context for the interpretation of *TTN* variants, as population-specific allele frequencies can influence variant classification, particularly in a gene as large and complex as *TTN* [62,63]. For instance, Haggerty et al. (2019) found that individuals of European ancestry may have an increased predisposition to carry *TTN* variants associated with dilated cardiomyopathy, whereas individuals of African ancestry showed a lower correlation between *TTN* variants and dilated cardiomyopathy [64]. Similarly, Jordan et al. (2023) found that African ancestry individuals diagnosed with DCM harbor fewer clinically actionable variants than their European counterparts, reflecting differences in genomic architecture and the lack of adequate ancestral representation in genomic reference datasets [65].

Taken together, our findings suggest that *TTN* may contribute to a broader spectrum of cardiac phenotypes than traditionally recognized, including cardiomyopathies, primary electrical disorders, and conduction disease. However, the functional impact of the variants identified in this cohort cannot be determined without experimental validation and familial segregation analyses. In addition, although this study focuses specifically on TTN, the clinical presentation of some patients may be partially or fully influenced by pathogenic variants in other cardiovascular genes included in the sequencing panel. Therefore, a comprehensive genetic evaluation of these additional genes in the 21 patients is necessary to improve phenotype interpretation and contextualize the contribution of TTN variants. These limitations emphasize the need for future studies to clarify whether specific *TTN* variants modulate structural and electrical traits in different clinical contexts.

The ancestry profiles of the individuals, predominantly Native American, did not reveal a clear relationship with variant distribution but did highlight the limited representation of Latin American genomes in public databases. Incorporating ancestry information, such as the NAM predominance observed in this cohort, may provide a more equitable framework for variant curation and support personalized genomic medicine that integrates both genetic variation and ancestry composition to improve diagnostic accuracy and clinical decision-making.

In summary, this study provides an initial framework for characterizing *TTN* variation in an underrepresented population and outlines key directions for future research, including analyses in larger cohorts and functional studies to improve variant interpretation.

## 4. Materials and Methods

### 4.1. Study Cohort

Sixty patients were recruited by cardiologists and electrophysiologists from clinical centers in Quito, Ecuador. Ages ranged from 9 days to 70 years. Eligible participants were individuals with a previously confirmed clinical diagnosis of a hereditary cardiac condition, totaling 37 distinct diagnoses. The cohort comprised 22 females and 38 males, all Ecuadorian by birth with both parents of Ecuadorian origin. Each participant provided detailed clinical and family histories, including information on relatives affected by cardiac disease. Recruitment followed convenience sampling among individuals who met the inclusion criteria and provided written informed consent.

### 4.2. Collection of Peripheral Blood and DNA Extraction

Peripheral blood samples were collected in EDTA tubes following written informed consent. Genomic DNA was extracted using the PureLink^TM^ Genomic DNA Mini Kit (Thermo Fisher, Waltham, MA, USA), following the manufacturer’s protocol. DNA concentration and quality were assessed using spectrophotometric and fluorometric methods prior to sequencing.

### 4.3. Next-Generation Sequencing (NGS)

All genomic procedures were performed at the Centro de Investigación Genética y Genómica (CIGG), Universidad UTE. Targeted NGS was conducted using the TruSight^TM^ Cardio Kit (Illumina, San Diego, CA, USA), which includes 174 genes (575 kb cumulative target region) associated with 17 inherited cardiac disorders. Library preparation followed the manufacturer’s instructions, and sequencing was carried out on the MiSeq™ System (Illumina, San Diego, CA, USA) using paired-end chemistry.

### 4.4. NGS Data Analysis

Bioinformatic analyses were performed on the Illumina BaseSpace^®^ (8 August 2025) cloud platform using the standard genomic sequencing pipeline. FASTQ files were automatically generated, and sequence reads were aligned to the human reference (GCRCh38) using DRAGEN Enrichment software v3.10.4 (Illumina, San Diego, CA, USA). Variant calling and annotation were performed using the Variant Interpreter platform (Illumina). The resulting variant data were exported to Excel for further visualization, filtering, and *TTN*-specific analysis.

### 4.5. NGS Filtration

Initial variants obtained from NGS were subjected to quality control filters, including evaluation of read depth, quality scores, and call confidence metrics. Variants that met the quality thresholds were flagged as “PASS”. To further prioritize variants with potential biological relevance, a second filtering stage was implemented based on variant consequence. Synonymous variants and intronic variants not predicted to affect canonical splice sites were removed, as these changes are unlikely to alter protein structure or function. All variants reported in ClinVar as benign or likely benign were excluded. The remaining variants were further evaluated according to the American College of Medical Genetics and Genomics (ACMG) criteria, including population frequency, in silico predictions, and previously reported information when available.

### 4.6. Variant Classification

All *TTN* variants were classified following the American College of Medical Genetics and Genomics (ACMG) and the Association for Molecular Pathology (AMP) guidelines [66], integrating gene-specific recommendations established by ClinGen [67].

The bioinformatic analyses were performed using the Franklin^®^ variant interpretation platform (Genoox, Tel Aviv, Israel; 15 October 2025). Evidence sources included: clinical databases, such as ClinVar, literature review, the Genome Aggregation Database (gnomAD v3.1), and in-silico prediction software tools including Revel v2.80, AlphaMissense, FATHMM v2.3, MutationAssessor, SIFT v1.2, MutationTaster, DANN, MetaLR, PrimateAI, and BayesDel.

Variants were categorized according to ACMG/AMP evidence codes. Those meeting strong or moderate pathogenic criteria were classified as likely pathogenic, whereas variants with insufficient or conflicting evidence were considered variants of uncertain significance (VUS). Synonymous, non-coding, and intronic variants classified as benign or likely benign were excluded from downstream analyses.

### 4.7. Genetic Ancestry Determination

Genetic ancestry was determined using Ancestry Informative Markers (AIMs) based on insertion/deletion (InDel) polymorphisms, following the protocol by Zambrano et al. (2019) [51]. Capillary electrophoresis and fragment analysis were performed using a 3500 Genetic Analyzer from Applied Biosystems (Waltham, MA, USA).

Ancestry inference analyses were conducted using STRUCTURE v2.3.4, comparing genotypic data from 46 AIMS-InDels markers against reference populations from the HGDP-CEPH panel (Africans, Europeans, and Native Americans; subset H952) [68,69]. The analysis employed an admixture model “Use population information to test for migrants”, with a burn-in length of 10,000, and 10,000 Markov Chain Monte Carlo (MCMC) iterations. The number of clusters (K) was evaluated from K = 1 to K = 3, and the results were visualized as triangular plots generated using STRUCTURE, to illustrate individual ancestry proportions and population admixture patterns.

### 4.8. Ethics Statement

The present study was evaluated and approved by the Human Research Ethics Committee (CEISH) of UTE University (CEISH-2021-016). The project was conducted in compliance with local legislation and institutional requirements. Written informed consent was obtained from the participants or the participants’ legal guardians/next of kin.

## Figures and Tables

**Figure 1 ijms-26-11896-f001:**
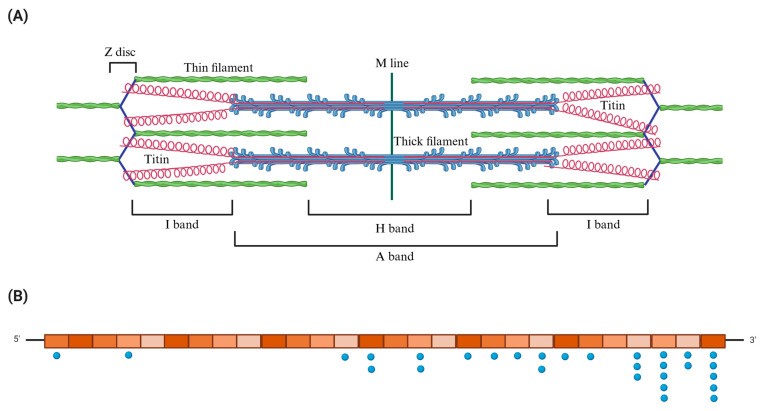
Structural representation of the sarcomere and schematic organization of the *TTN* gene with mapped variants. The upper panel (**A**) depicts the sarcomere architecture, highlighting the spatial distribution of titin in relation to the major contractile elements. Thick filaments (myosin) are shown in blue, thin filaments (actin) in green, and the titin filaments are highlighted in pink, spanning from the Z-disc to the M-line. The lower panel (**B**) presents a schematic representation of the *TTN* gene, divided into 28 proportional segments, each corresponding to 13 exons of the canonical transcript. Blue dots mark the locations of variants, distributed across different exon groups along the 5′-3′ structure of the gene. Created in https://BioRender.com.

**Figure 2 ijms-26-11896-f002:**
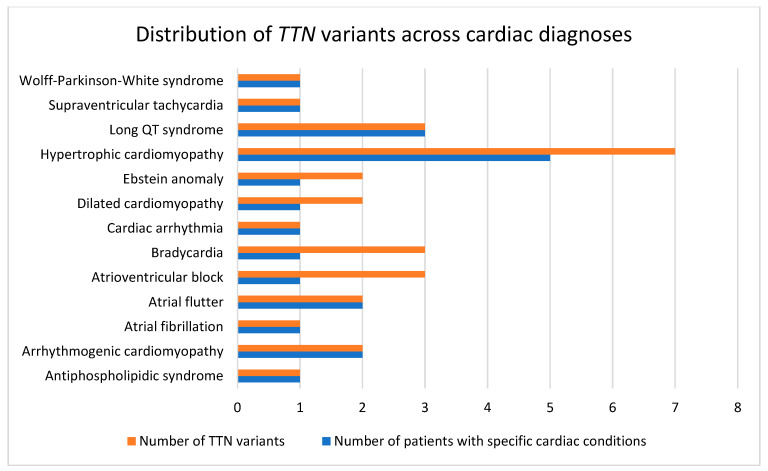
Distribution of *TTN* variants across cardiac diagnoses in the study. The horizontal bar chart compares the number of *TTN* variants identified (orange bars) with the number of patients diagnosed (blue bars) for each cardiac condition observed in the study cohort.

**Figure 3 ijms-26-11896-f003:**
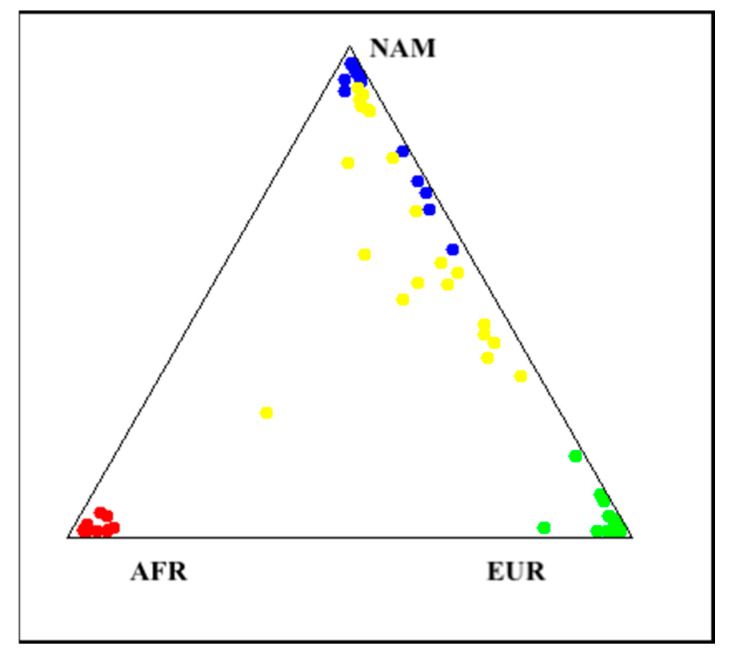
Genetic ancestry proportions in the cohort. The figure plot depicts the genetic ancestry composition of the Ecuadorian patients included in the project (yellow). Genetic ancestry determination was based on 46 AIM–InDel markers. Each point represents an individual sample, positioned according to its estimated proportion of ancestry from three reference populations: African ancestry (red), Native American ancestry (blue), and European ancestry (green).

**Table 1 ijms-26-11896-t001:** *TTN* variants identified and its ACMG/AMP classification.

SAMPLE	Diagnosis	Exon	HGVSC	HGVSP	Consequence	Zygosity	rs	Allele Frequency	ACMG/AMP Classification
ADN001	Long QT syndrome	358/363	c.103957C>T	p.(Arg34653Cys)	missense_variant	Heterozygous	rs773002407	<0.01%	PM2
ADN008	Hypertrophic cardiomyopathy	326/363	c.74135T>C	p.(Leu24712Pro)	missense_variant	Heterozygous	rs570875618	<0.01%	PM2
		284/363	c.55195C>T	p.(Pro18399Ser)	missense_variant	Heterozygous	rs774591174	<0.01%	PM2
		244/363	c.44986C>T	p.(Arg14996Cys)	missense_variant	Heterozygous	rs765602798	<0.01%	PM2
									BP6
ADN013	Long QT syndrome	358/363	c.101959C>T	p.(His33987Tyr)	missense_variant	Heterozygous	no rs	N/A	PM2
ADN014	Hypertrophic cardiomyopathy	358/363	c.102827G>A	p.(Arg34276Gln)	missense_variant	Heterozygous	rs199932621	0.02%	PM2
									BP6
ADN023	Arrhythmogenic cardiomyopathy	325/363	c.69638G>A	p.(Arg23213His)	missense_variant	Heterozygous	rs374883884	0.02%	PM2
ADN037	Antiphospholipidic syndrome	326/363	c.85286T>A	p.(Val28429Glu)	missense_variant	Heterozygous	rs747216991	0.01%	PM2
ADN050	Dilated cardiomyopathy	267/363	c.50293A>T	p.(Lys16765Ter)	stop_gained	Heterozygous	no rs	N/A	PVS1
									PM2
		326/363	c.78328A>G	p.(Thr26110Ala)	missense_variant	Heterozygous	no rs	N/A	PM2
ADN058	Atrial Flutter	326/363	c.73847G>A	p.(Arg24616Gln)	missense_variant	Heterozygous	rs201694149	0.04%	PM2
									BP6
ADN093	Arrhythmogenic cardiomyopathy	344/363	c.95540G>A	p.(Arg31847His)	missense_variant	Heterozygous	rs771179752	0.07%	PM2
									PP3
ADN097	Atrial flutter	170/363	c.36418C>T	p.(Pro12140Ser)	missense_variant	Heterozygous	rs1225570991	<0.01%	PM2
									BP4
ADN099	Hypertrophic cardiomyopathy	344/363	c.95464G>A	p.(Glu31822Lys)	missense_variant	Heterozygous	rs986919717	<0.01%	PM2
ADN100	Wolff-Parkinson-White syndrome	169/363	c.36281T>C	p.(Val12094Ala)	missense_variant, splice_region_variant	Heterozygous	no rs	N/A	PM2
									BP4
ADN101	Atrial fibrillation	204/363	c.39250G>A	p.(Val13084Met)	missense_variant	Heterozygous	rs72650062	0%	PM2
									BP4
ADN103	Supraventricular tachycardia	358/363	c.102839C>T	p.(Thr34280Ile)	missense_variant	Heterozygous	rs886042505	<0.01%	PM2
ADN106	Cardiac arrhythmia	272/363	c.51546C>A	p.(Tyr17182Ter)	stop_gained	Heterozygous	no rs	N/A	PVS1
									PM2
ADN116	Hypertrophic cardiomyopathy	3/363	c.271A>G	p.(Ser91Gly)	missense_variant	Heterozygous	no rs	N/A	PM2
ADN117	Hypertrophic cardiomyopathy	253/363	c.47498T>A	p.(Val15833Glu)	missense_variant	Heterozygous	rs758495958	<0.01%	PM2
									PP3
ADN120	Long QT syndrome	170/363	c.36418C>T	p.(Pro12140Ser)	missense_variant	Heterozygous	rs1225570991	<0.01%	PM2
									BP4
ADN130	Atrioventricular block	313/363	c.65672C>T	p.(Pro21891Leu)	missense_variant	Heterozygous	rs397517662	0.02%	PM2
									PP3
		231/363	c.42625T>A	p.(Ser14209Thr)	missense_variant	Heterozygous	no rs	N/A	PM2
		204/363	c.39250G>A	p.(Val13084Met)	missense_variant	Heterozygous	rs72650062	0%	PM2
									BP4
ADN146	Bradycardia	358/363	c.101824C>T	p.(Pro33942Ser)	missense_variant	Heterozygous	no rs	N/A	PM2
		326/363	c.74935T>C	p.(Tyr24979His)	missense_variant	Heterozygous	rs1364199300	<0.01%	PM2
									PP3
		315/363	c.66313G>T	p.(Val22105Phe)	missense_variant	Heterozygous	no rs	N/A	PM2
ADN150	Ebstein anomaly	287/363	c.55730T>C	p.(Ile18577Thr)	missense_variant, splice_region_variant	Heterozygous	no rs	N/A	PM2
		49/363	c.14302G>A	p.(Gly4768Ser)	missense_variant	Heterozygous	rs727503652	<0.01%	PM2
									PP3
									BP6

## Data Availability

The original contributions presented in this study are included in the article/Appendix A. Further inquiries can be directed to the corresponding author.

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
