# Peer review of "Spectrum and Clinical Interpretation of *TTN* Variants in Ecuadorian Patients with Heart Disease: Insights into VUS and Likely Pathogenic Variants"

_ijms, 2025, doi:10.3390/ijms262411896_

Round 1

Reviewer 1 Report

Comments and Suggestions for Authors

This article is an excellent article about TTN variants in Ecuadorian patients.The manuscript entitled “Spectrum and Clinical Interpretation of TTN Variants in Ecuadorian Patients with Heart Disease: Insights into VUS and Likely Pathogenic Variants" is about TTN variants in Ecuadorian patients. 

Minor point

Please make a filtration section in a method and explain about quality control filtering and filtering based on variant consequence.

Author Response

This article is an excellent article about TTN variants in Ecuadorian patients.The manuscript entitled “Spectrum and Clinical Interpretation of TTN Variants in Ecuadorian Patients with Heart Disease: Insights into VUS and Likely Pathogenic Variants" is about TTN variants in Ecuadorian patients. 

Thank you for the positive assessment and helpful comments. The suggested modifications have been implemented, and all changes are highlighted in yellow in the revised manuscript.

Minor point

Please make a filtration section in a method and explain about quality control filtering and filtering based on variant consequence.

Authors’ response: Thank you for your comment. We have included a filtration section, explaining about quality control filtering, as suggested.

Reviewer 2 Report

Comments and Suggestions for Authors

The Introduction section is too short and the Discussion section is overloaded with information that is not related to the results obtained.

It's not always clear whether the authors are referring to literary data or their own

It is necessary to move the brief description of the TTN protein from the Discussion to the Introduction section because information about its domains appears for the first time without explanation in Figure 1 in the Results section.

It would also be logical to justify the use of the definition of genetic ancestry in the Introduction.

The Figure 1 needs correction.

1) Title: This isn't a gene, but RNA or cDNA.

2) Formatting part of the figure as a callout is unreasonably. I think it would make more sense to make two parts of the figure (for example, A and B).

3) In its current form, figure doesn't correspond to what's written in the text (This protein spans the Z-disk, I-band, A-band, and M-line regions – Line 192). In this figure, TTN protein is in I-band only

In Table 1, an abbreviation Dx is without explanation

I suppose adding allele frequencies (according GnomAD) to column with rs ID of the Table will make the results clearer.

Line 151: TTN variants were detected in patients across multiple cardiac phenotypes

“Above mentioned” should be added to clarify text.

In Figure 3, red dots had been described twice, yellow and green dots have not description

There is no information on whether pathogenic variants were detected in other 173 cardiac genes studied in the patients described (except ADN001).

Line 219: Therefore, missense mutations could be involved in a lower interaction with sarcomeric related proteins, which leads to cardiomyopathies like DCM.

I think it would be more correct to write "may influence interaction"

Line 217 If myosin binding sites are mentioned, how are the variants located relative to them? Or why is this information given in the Discussion section?

Line 228: These diverse clinical phenotypes could be related to the different expression levels of wild types in each individual, which could interfere with the sarcomere assembly and its interaction with non sarcomeric proteins

What is meant by wild types?

Line 233: TTN Missense variants have been associated with impaired Titin protein that leads to defects in sarcomere assembly and function (5).

Do you mean some described missense variants or something else?

In Line 246, what is table x?

Lines 264-279 contain information about sex differences in heart diseases, but any information about sex of described patients is absent in the Table. Why do authors need this?

Line 289: In this cohort, seven heterozygous missense variants (p.Leu24712Pro, p.Pro18399Ser, p.Arg14996Cys, p.Arg34276Gln, p.Glu31822Lys, p.Ser91Gly, and p.Val15833Glu) were identified in the TTN gene, distributed between exons 3 and 358.

It is unclear what this is all about; 29 variants were mentioned previously.

Line 399: suggesting TTN as a new candidate gene for familial (68).

Word is missing?

Lines 421-432: It is not clear why this information was inserted, since it does not confirm “notable patterns that may contribute to understanding both population origins and the limited representation of Latin American genetic diversity in global database”

I think the authors should not try to make a literature review out of the Discussion section, but instead focus on discussing their results.

Line 450: In conclusion, these results highlight the importance of ancestry-informed interpretation of TTN variants

It is unclear why the importance of ancestry-informed interpreting of TTN variants was concluded, as the authors state that some genetic variants are common across populations, and for undescribed variants, functional and segregation analysis is needed before the role of these variants in pathogenesis specifically in NAM can be discussed.

Author Response

We appreciate the thoughtful comments and suggestions provided. All recommended changes have been incorporated, and the corresponding revisions are highlighted in yellow in the updated manuscript.

The Introduction section is too short and the Discussion section is overloaded with information that is not related to the results obtained.

Authors’ response: Thank you for your comment. More information has been added to the introduction section. The Discussion section has been reorganized to focus strictly on the results obtained. Content not directly supported by our findings was removed or condensed, ensuring a clearer and more coherent interpretation aligned with the data.

It's not always clear whether the authors are referring to literary data or their own.

Authors’ response: Thank you for your comment. All the sections were reviewed.

It is necessary to move the brief description of the TTN protein from the Discussion to the Introduction section because information about its domains appears for the first time without explanation in Figure 1 in the Results section.

Authors’ response: Thank you for your comment. The information about TTN protein has been moved from the Discussion to the Introduction section.

It would also be logical to justify the use of the definition of genetic ancestry in the Introduction.

Authors’ response: Thank you for your comment. The reason behind including genetic ancestry was added in the introduction.

The Figure 1 needs correction.

Title: This isn't a gene, but RNA or cDNA.

Formatting part of the figure as a callout is unreasonably. I think it would make more sense to make two parts of the figure (for example, A and B).

Authors’ response: Thank you for your comment. We agree. The figure has been redesigned into two separate panels, in which the protein schematic is depicted in panel A, and the gene in panel B, as it describes where the genetic variants have been identified. Furthermore, the figure legend has been updated accordingly.

In its current form, figure doesn't correspond to what's written in the text (This protein spans the Z-disk, I-band, A-band, and M-line regions – Line 192). In this figure, TTN protein is in I-band only.

Authors’ response: Thank you for your comment. We have corrected this discrepancy. The updated sarcomere diagram now accurately depicts titin spanning all major structural regions: the Z-disc, I-band, A-band, and M-line, consistent with the manuscript description.

In Table 1, an abbreviation Dx is without explanation

Authors’ response: Thank you for your comment. We have corrected this.

I suppose adding allele frequencies (according GnomAD) to column with rs ID of the Table will make the results clearer.

Authors’ response: Thank you for your comment. We have included, in the table, the allele frequencies (according to GnomAD) as suggested.

Line 151: TTN variants were detected in patients across multiple cardiac phenotypes. “Above mentioned” should be added to clarify text.

Authors’ response: Thank you for your comment. “Above mentioned” has been included as suggested.

In Figure 3, red dots had been described twice, yellow and green dots have not description.

Authors’ response: Thank you for your observation. The Figure 3 caption has been updated to describe all the dots correctly.

There is no information on whether pathogenic variants were detected in other 173 cardiac genes studied in the patients described (except ADN001).

Authors’ response: Thank you for your comment. The primary focus of our manuscript is the TTN gene and its potential association with cardiovascular. Although the sequencing panel included 174 cardiac-related genes, the analysis and interpretation of variants identified in the remaining genes fall outside the scope of the present study. These additional genes, and their corresponding variants, have been studied and are being reported in separate manuscripts dedicated to those specific conditions and genetic findings.

For the current article, we centered the analysis on TTN, given its biological relevance, the diversity of its variant spectrum, and its clinical importance in cardiomyopathies.

Line 219: Therefore, missense mutations could be involved in a lower interaction with sarcomeric related proteins, which leads to cardiomyopathies like DCM.

I think it would be more correct to write "may influence interaction"

Authors’ response: Thank you for your comment. We have modified this sentence as suggested.

Line 217 If myosin binding sites are mentioned, how are the variants located relative to them? Or why is this information given in the Discussion section?

Authors’ response: Thank you for your comment. We have included a paragraph describing that exons located at the C-terminal region of the A-band are mainly involved in the maintenance of thick filament stability through the binding to the myosin tail, and that this interaction could be disrupted in the presence of TTN missense variants and promote cardiac phenotypes. Moreover, we added the C-terminal exons with missense variants found in this study and their potential implication in DCM.

This statement is given to explain how the location of the TTN missense variants could be involved in defects of TTN-myosin interactions.

Line 228: These diverse clinical phenotypes could be related to the different expression levels of wild types in each individual, which could interfere with the sarcomere assembly and its interaction with non sarcomeric proteins

What is meant by wild types?

Authors’ response: Thank you for your observation. The wording has been corrected in the revised manuscript. The term “wild types” has been replaced with “wild-type TTN allele” to accurately reflect the expression of the non-mutated allele.

Line 233: TTN Missense variants have been associated with impaired Titin protein that leads to defects in sarcomere assembly and function (5).

Authors’ response: Thank you for your comment. Effectively, we are referring to TTN missense variants that can be involved in defects in sarcomere assembly. Moreover, we have added two references that describe the type of missense variant in the TTN gene and the associated cardiac phenotype. (https://doi.org/10.1016/j.ijcard.2013.11.037; https://doi.org/10.1016/j.cjca.2018.07.473)

Do you mean some described missense variants or something else?

In Line 246, what is table x?

Authors’ response: Thank you for your observation. We have corrected this mistake and placed the correct statement.

Lines 264-279 contain information about sex differences in heart diseases, but any information about sex of described patients is absent in the Table. Why do authors need this?

Authors’ response: Gracias por el comentario. Dado que la información sobre el sexo se incluye en la sección de Resultados, creemos que una breve discusión de las diferencias relacionadas con el sexo es relevante para contextualizar la interpretación de las variantes. El texto se ha revisado para mantener una conexión más clara con nuestros datos.

Line 289: In this cohort, seven heterozygous missense variants (p.Leu24712Pro, p.Pro18399Ser, p.Arg14996Cys, p.Arg34276Gln, p.Glu31822Lys, p.Ser91Gly, and p.Val15833Glu) were identified in the TTN gene, distributed between exons 3 and 358.

It is unclear what this is all about; 29 variants were mentioned previously.

Authors’ response: Thank you for your comment. We have clarified this section. The text now specifies that the seven heterozygous missense variants represent a subset of the 29 TTN variants identified in the cohort and were specifically observed in patients diagnosed with DCM, resolving the previous ambiguity.

Line 399: suggesting TTN as a new candidate gene for familial (68).

Word is missing?

Authors’ response: Thank you for your comment. We have corrected the incomplete sentence by specifying the missing term. The revised text now reads “familial atrioventricular block,” and the updated wording is reflected in the manuscript.

Lines 421-432: It is not clear why this information was inserted, since it does not confirm “notable patterns that may contribute to understanding both population origins and the limited representation of Latin American genetic diversity in global database”

I think the authors should not try to make a literature review out of the Discussion section, but instead focus on discussing their results.

Authors’ response: Thank you for the comment. The ancestry paragraph has been revised to remove unrelated background information and now focuses exclusively on the patterns observed in our cohort.

Line 450: In conclusion, these results highlight the importance of ancestry-informed interpretation of TTN variants. It is unclear why the importance of ancestry-informed interpreting of TTN variants was concluded, as the authors state that some genetic variants are common across populations, and for undescribed variants, functional and segregation analysis is needed before the role of these variants in pathogenesis specifically in NAM can be discussed.

Authors’ response: Thank you for the comment. The conclusion was revised to avoid overinterpretation; ancestry is now presented only as contextual information, not as evidence of pathogenicity.

Round 2

Reviewer 2 Report

Comments and Suggestions for Authors

Figure 1

It is not entirely clear from the figure where the TTN protein is located; perhaps it would be worth indicating that it is marked in pink, for example.

Authors’ response:

Thank you for your comment. The primary focus of our manuscript is the TTN gene and its potential association with cardiovascular. Although the sequencing panel included 174 cardiac-related genes, the analysis and interpretation of variants identified in the remaining genes fall outside the scope of the present study. These additional genes, and their corresponding variants, have been studied and are being reported in separate manuscripts dedicated to those specific conditions and genetic findings.

For the current article, we centered the analysis on TTN, given its biological relevance, the diversity of its variant spectrum, and its clinical importance in cardiomyopathies.

This is entirely reasonable, but in this case, it should be noted that the clinical picture observed in the described patients with variants in the TTN gene may be partially or completely explained by pathogenic variants in other genes associated with cardiovascular disease. I believe it is necessary to emphasize in the "Limitations of the Study" section the need for genetic analysis of variants in other genes associated with cardiovascular disease in 21 patients described.

Line 412

Duplication of the word "similarly"

Author Response

Thank you for the constructive comments and valuable suggestions. All updates have been incorporated and highlighted in yellow, and the English has been carefully revised using tracked changes.

Figure 1

It is not entirely clear from the figure where the TTN protein is located; perhaps it would be worth indicating that it is marked in pink, for example.

Authors’ response: Thank you for your comment. We have highlighted titin in pink and updated the legend to clearly indicate its location in the figure.

Authors’ response:

Thank you for your comment. The primary focus of our manuscript is the TTN gene and its potential association with cardiovascular. Although the sequencing panel included 174 cardiac-related genes, the analysis and interpretation of variants identified in the remaining genes fall outside the scope of the present study. These additional genes, and their corresponding variants, have been studied and are being reported in separate manuscripts dedicated to those specific conditions and genetic findings.

For the current article, we centered the analysis on TTN, given its biological relevance, the diversity of its variant spectrum, and its clinical importance in cardiomyopathies.

This is entirely reasonable, but in this case, it should be noted that the clinical picture observed in the described patients with variants in the TTN gene may be partially or completely explained by pathogenic variants in other genes associated with cardiovascular disease. I believe it is necessary to emphasize in the "Limitations of the Study" section the need for genetic analysis of variants in other genes associated with cardiovascular disease in 21 patients described.

Authors’ response: Thank you for this valuable comment. We have now clarified in the limitations paragraph that a comprehensive genetic evaluation of variants in other cardiovascular disease–associated genes in the 21 patients is necessary to better contextualize their clinical phenotypes.

Line 412

Duplication of the word "similarly"

Authors’ response: Thank you for noting this. The duplicated word has been corrected.
